# Technical note. On the importance of a three-dimensional approach for modelling the transport of neustic microplastics

Isabel Jalón-Rojas[1], Xiao-Hua Wang[1], Erick Fredj[2]

[1]The Sino-Australian Research Centre for Coastal Management, School of Physical, Environmental and Mathematical Sciences, UNSW Canberra, Canberra, 2610, Australia
[2]Jerusalem College of Technology, Jerusalem, Israel

*Correspondence to*: Isabel Jalón-Rojas (i.jalonrojas@unsw.edu.au; ijalonrojas@gmail.com)

**Abstract.** Understanding and estimating the distribution and transport of microplastics in marine environments has been recognized as a major global research issue. Most of the existing research on transport modelling has focused on low-dense particles floating in surface waters, using a 2D Lagrangian approach and ignoring the vertical displacement of particles. In this work, we evaluate to what extent the vertical movement of particles within surface waters by mixing processes may affect the horizontal transport and fate of microplastics. The aim is to determinate whether a 2D approach is sufficient for the accurate modelling of neustic-microplastics transport or a 3D approach is necessary. For this purpose, we compare visually and statistically the microplastics transport patterns of three simulations in a coastal system: one using a 2D approach; and two using a 3D approach with weak and strong vertical turbulence, respectively. The 2D simulation roughly reproduced the transport and accumulation patterns, but accurate results required a 3D approach. This was particularly important for strong vertical turbulence and regions characterized by strong vertical current shear. Moreover, a 2D approach can lead to errors in the results even with negligible turbulence due to simplifications in the velocity field. A 3D modelling approach is therefore key to an accurate estimation and prediction of microplastics distribution in coastal systems, and consequently for planning mitigation and cleaning programs.

## 1 Introduction

Marine plastic debris is of increasing concern because of its persistence, toxicological properties, and effects on marine ecosystems, wildlife, and humans (Lithner et al., 2011; Rochman et al., 2013). In particular, microplastics (<5 mm) are the most abundant and potentially hazardous plastic items in marine environments (Andrady, 2011). Microplastics pollution has been documented throughout the world and marine habitats (Eriksen et al., 2014; Galgani et al., 2015), in the surface and subsurface water column, on the sea floor, along coastlines and in the polar regions. Microplastics can also accumulate in marine organisms at different trophic levels (Carbery et al, 2018).

Understanding, estimating and predicting the distribution and transport of microplastics is a key step in addressing this global issue. This is a complex problem that requires in-situ observations and numerical modelling. Numerous studies have used Lagrangian particle-tracking models to assess the sources, pathways and sinks of microplastics in marine environments,

especially at ocean and regional scales (e.g. Wakata and Sugimori, 1990; Isobe et al., 2009; Ebbesmeyer et al., 2012; Lebreton et al., 2012; Critchell and Lambrechts, 2016; Carlson et al., 2017; Liubartseva et al., 2018). These models typically consider particles moving in surface waters, and therefore use a two-dimensional approach based on surface or vertically averaged current velocities. However, microplastics can move through the water column for different reasons (Zhang, 2017) – neutral

and negative buoyancy, vertical mixing processes, upwelling/downwelling processes – and a three-dimensional approach could be necessary. In the case of positive buoyant microplastics, observations revealed that they normally remain near the surface but can move within a layer of depth up to 5m due to hydrodynamic mixing processes (Reisser et al., 2015, Kooi et al., 2016). This vertical movement may affect the horizontal trajectories of microplastics in stratified systems, but this impact has not yet been explored.

The aim of this work is to evaluate the accuracy of a two-dimensional approach for modelling the transport of positive-buoyant neustic microplastics under conditions of low and high vertical turbulence in coastal shallow environments. We implement a Lagrangian particle-tracking model to compare the trajectories and fates of microplastics from 2D simulations with those from 3D (used as reference solutions), using Jervis Bay (SE Australia) as a natural laboratory. This comparison is performed both visually and quantitatively using probability-density maps and coastal connectivity analysis. This work also provides a first

insight into the pertinence of a 3D approach for modelling the transport of negative-buoyant microplastics subject to sinking and other physical processes involving vertical transport.

## 2 Methods

### 2.1 Lagrangian Particle-Tracking Model

The Particle Tracking and Analysis TOolbox (PaTATO, Fredj et al., 2016) was used for modelling the advection-diffusion of
buoyant particles in 2D and 3D. Advective and diffusive displacement determine the trajectories of particles as:

$$\mathbf{dX}(t) = \mathbf{U}dt + \mathbf{R}\sqrt{2Kdt}, \tag{1}$$

The particle displacement $\mathbf{dX}$=(dX, dY, dZ) is given by the flow velocity U=(U,V,W), which can provided by diverse hydrodynamic models (e.g. POM, ROMS), and a stochastic term related to the to the dispersion coefficients $\mathbf{K}$=($K_{h,x}$, $K_{h,y}$, $K_{h,z}$). $\mathbf{R}$=($R_x$,$R_y$,$R_z$) represents white-noise random walks with an average and standard deviation of 0.0 and 1.0. Particles can
beach when their positions are inside the land domain.

### 2.1 Model settings

The Lagrangian particle-tracking model was implemented in Jervis Bay (SE Australia, Fig. 1.a) in two and three dimensions, in order to compare the results of two approaches. This semi-enclosed bay is 8 km wide, 15 km long and 15 m deep on average. We used hydrodynamic model results from the Princeton Ocean Model (POM) as inputs. In particular, we used the
hydrodynamics data from 24 June to 11 July 1998, as the model was validated using observation of currents, temperature,

salinity and water level during this period, obtaining a very good fit (Wang and Symonds, 1999; Sun et al., 2017; Liao and Wang, 2018). The POM model spatial resolution is 500 m around the bay, and the temporal resolution is 12 s. It uses a total of twenty-one sigma levels in the z-direction, with finer layers near the surface and bottom. During this simulation period, Jervis Bay was characterized by its typical circulation pattern: clockwise and anticlockwise circulation in the northern and southern regions, respectively. The flow exchange through the entrance was highly stratified, with near-surface inflow on the southern side and deeper outflow on the northern side (Fig 1.b). Vertical currents were 3-4 order of magnitude lower than horizontal currents (Fig 1.b). However, the aim of this work is not to discuss the typical patterns of microplastics transport in Jervis Bay; it is a case study to explore the implications of a 2D approach on the simulation accuracy of neustic-microplastics transport in coastal shallow waters. The reader is referred to Sun et al. (2017) for more details on the hydrodynamic model settings.

We implemented three model simulations to evaluate the accuracy of a 2D approach for modelling microplastics floating in surface waters:

a)      2D approach

b)      3D approach with weak vertical turbulence

c)      3D approach with strong vertical turbulence

The only processes involved in the transport of microplastics in the three simulations were advection, diffusion and beaching, as in most of the existing 2D models for the transport of microplastics. The 2D approach used surface currents in the first sigma level of the POM domain (layer thickness around 0.08 m at the inner bay and 0.3 m at the mouth). The reference 3D approaches used the whole 3D velocity dataset. Weak and strong vertical turbulence were defined by low ($10^{-5}$ $m^2s^{-1}$) and high ($10^{-4}$ $m^2s^{-1}$) values of the vertical diffusivity coefficients typical of marine systems (Talley et al., 2011). With weak turbulence, particles remained near the surface during the whole simulation (standard deviation of particles depths, σ, equals to 0.7 m), whereas strong turbulent conditions induced high vertical displacements of particles (σ equals to 2.5 m). In order to compare only the impact of vertical turbulence on the horizontal trajectories, the effect of the horizontal diffusivity on the trajectory of each particle at each time step was considered identical in all three scenarios. Therefore, vertical displacements will be the only possible cause of the potential differences between the horizontal transport patterns of the three different approaches. In order to avoid the random behaviour of turbulent dispersion having an impact on the comparison of simulations, the same turbulent horizontal and vertical displacements were assigned to each particle at each time step for all scenarios.

All the simulations were seeded at 18 locations (Fig 1) covering the whole coast of Jervis Bay in order to analyse the bay connectivity. Twenty particles per hour were released at each seeding site in surface waters during three days from 26 June 1998, a total of 25,920 particles per simulation. A sensitivity analysis of model results to the number of particles is provided in the Supplementary Material to demonstrate that results were not affected by this parameter. Simulations were run for five days, as most of particles reached the coast during this period.

## 2.3 Probability distributions and connectivity analysis

To evaluate the prediction accuracy of a 2D approach in modelling the transport of neustic microplastics, we compared the trajectories and fates of microplastics obtained from 2D and 3D simulations using three methods: (1) visual and descriptive comparison of the resulting trajectories and fate; (2) probability-density maps; and (3) coastal connectivity analysis.

- **Probability-density maps**

This analysis facilitates visualizing and understanding particle trajectories by binning particle positions into histograms. It calculates the probability that a particle moves from one location to another over a time interval $\tau$ by counting the number of particles per bin and then normalizing by the total number of particles. Bins were defined by the grid of the hydrodynamic

model ($500 \times 500$m). The result is a probability map of particle density.

- **Coastal connectivity analysis**

Coastal connectivity $C_{ji}(\tau)$ is the probability that a particle leaving a source site j arrives a destination site i over the time interval $\tau$, calculated in the same way as for the probability-density maps, using the site locations shown in Fig. 1 as source and destination locations. Destinations are rectangular areas of 1.5 km$^2$ centred on the site locations of Fig. 1. $C_{ji}(\tau)$ was thus

calculated for five time intervals $\tau$ = 1, 2, 3, 4 and 5 days. This results in five $18 \times 18$ connectivity matrices that describe the probability of microplastics particles being transported between different sites along the whole Jervis Bay coast. Readers should refer to van Sebille et al. (2018) and Mitarai et al. (2009) for a detailed description and application of both probability-density maps and connectivity analysis.

## 3 Results and Discussion

The trajectories and fates of microplastics released at the different source locations (Fig. 1) were plotted for the three scenarios to gain a first insight into the differences between the 2D and 3D approaches. Figure 2.I illustrates these results for microplastic particles released at sources 2 and 9. A first comparison of the three scenarios shows differences in the dispersion patterns of particles between the 2D and 3D simulations. In general, particles released at a given source reached similar coastal regions in all three approaches. However, the particles seemed to have a higher horizontal spread in the 3D approach (Fig 2.I.B-C),

especially with strong vertical turbulence (Fig. 2.I.C). In the case of particles released near the entrance (source 1), the vertical turbulent displacement can induce either a surface inflow or a deep outflow, so some particles could leave the bay in outflow deep currents with strong turbulence (Fig. 2.I.C), whereas all the particles stayed in the bay in the 2D approach (Fig. 2.I.A) and the 3D approach with weak vertical turbulence (Fig. 2.I.B). Even in these two last cases, in which particles barely moved vertically, the dispersion patterns showed some differences. Whereas most of particles reached the coastal region in the 2D

simulation (Fig. 2.I.A), some particles remained in suspension in the bay after 5 days of simulation in the 3D approach (Fig. 2.I.B). Differences between 2D and 3D weak-turbulence simulations were less evident for particles released in the inner bay, for example at source 9 (Fig. 2.I.A-B). In this case, most of the particles had accumulated in the inner bay by the end of the

simulation; with strong turbulence, some particles ended up in the middle of the bay (Fig. 2.I.C). However, this kind of analysis does not tell us whether most of the particle trajectories followed the same pattern in all three approaches, so that the visible differences did not show significant trends, or whether the visible differences showed the most common trends.

For a statistical quantification and comparison of the transport trends of the different approaches, we turn to probability density-maps; these shows the probability that a particle released at a given source moves to different regions of the bay. Figure 2.II shows the accumulation regions of particles released at sources 2 and 9 after five days of simulation. The accumulation patterns of the different approaches were not as different as suggested in Fig. 2.I, and the transport patterns were roughly similar for the 2D and 3D approaches. In all three scenarios, particles released at sources 2 and 9 accumulated mainly in the northern and eastern regions of the bay (Fig. 2.II). Nevertheless, there were clear differences between the 2D and 3D results. For particles released at source 9 in the inner bay with strong vertical turbulence (Fig. 2.II.C), these differences were significant. Whereas there was a significant accumulation spot of microplastics on the eastern coast of the bay for the 2D and 3D weak-turbulence approaches (dark grey, Fig.2.II.A-B), the particles fate was more spread out with strong vertical turbulence (Fig. 2.II.C). For particles released near the entrance (source 2), differences in the fate patterns appeared between the 2D and 3D approaches, even for weak vertical turbulence (Fig 2.II.A-B). In this case, there was a higher spread of particles in both 3D approaches.

For an overview of how the different modelling approaches influenced the transport patterns of neustic microplastics in the whole bay and as particles were released, Fig. 3 shows the percentage of particles traveling from and to the different coastal regions of the bay (18 sites in Fig. 1; see Section 2) for the three different scenarios, and simulation times from 1 to 5 days. This analysis confirmed most of conclusions reached above. First, both the 2D and 3D approaches provided similar transport patterns, but there were some clear differences. These differences appeared even when vertical turbulent dispersion was weak, although they were more evident with strong vertical turbulence. In particular, particle fates were concentrated over a shorter length of the coast in the 2D approach (fewer destinations for each source and higher probabilities of a specific destination, Fig. 3.A), whereas the particle spread was greater in the 3D approaches, especially for strong vertical turbulence (more destinations, with lower probabilities for each source; Fig. 3.B-C). These differences were more significant: (1) in sources near the entrance due to the higher vertical gradients of horizontal currents (see sources from 1–9, 17 and 18 in Fig. 3); and (2) for longer simulation times, as more particles had time to reach the coast, in particular from simulation times higher than 3 days (Fig. 3.III-IV).

In summary, the vertical transport of particles within surface layers due to mixing processes can affect the horizontal trajectories and fates of microplastics, particularly in systems with sharp horizontal velocity gradients. Although a 2D approach can predict general patterns, a 3D approach is recommended to improve the accuracy of the results, especially in the presence of strong turbulence. 3D simulations may even be necessary with weak vertical turbulence. This is because a 2D approach can cause errors in the results due to the use of velocities at a single sigma layer which represent different depths along the particle trajectory. This can lead to unphysical vertical movements and therefore to wrong horizontal patterns in very stratified systems. Vertically averaged current velocities could also lead to errors in the horizontal trajectories. In the 3D approach, the vertical resolution of the hydrodynamic model should be finer enough to represent the vertical current shear accurately.

All these findings may be transferable to (1) the other hydrodynamic conditions of this bay which are also dominated by baroclinic processes (coastal trapped waves, upwelling, cooling events; e.g. Wang and Symonds, 1999; Sun et al., 2017; Liao and Wang, 2018), (2) other stratified coastal systems such as estuaries characterized by a density circulation, but also to oceanic water characterized by vertical current shear induced by wind and wave-driven Ekman flow or density-driven processes, for

instance (Lund et al., 2015; Lanotte et al., 2016). Besides the microplastics, these results can also be applied for other floating "young" objects. A 2D approach may be sufficient when vertical current shear is negligible. In the present case study, the transfer of particle through the water column was mainly due to vertical dispersion, while vertical advection was negligible (see the impact of vertical advection on Supplementary Material 2). However, these results also highlight that the vertical movement of particles induced by other physical processes, such as upwelling, downwelling, wave enhancing vertical mixing,

and particle sinking (in the case of non-buoyant particles), could also affect the horizontal transport of microplastics, even in a higher degree, and a 3D approach could be mandatory. Some modelling studies consider the particle sinking in a 2D approach (Critchell and Lambrechts, 2016; Liubartseva et al., 2018), so the vertical current shear during the vertical transport is not taken into account. Further progress on microplastics modelling requires thus the development of three-dimensional models that consider the particle sinking, which in turn depends on particle physical properties (density, size, shape, Chubarenko et

al., 2016). This has been analysed in depth in our work (Jalon-Rojas et al., 2019).

In short, a 3D approach improves the simulation of the vertical position of particles in all turbulence conditions, which impact the predictions of the horizontal trajectories and fates of low-density neustic particles, especially in stratified systems. These results have important implications for the assessment and prediction of pollution hot spots in coastal systems, as well as for planning effective clean-up programs.

**Acknowledgment**

The authors thank Peter McIntyre for proofreading the article and the National Computational Infrastructure (NCI) at the Australian National University for the computational support. This is publication no. 63 of the SARCCM at UNSW Canberra.

**Author contribution**

Isabel Jalón-Rojas developed the study concept and performed the analysis under the supervision of Erick Fredj and Xiao Hua

Wang. The manuscript was drafted by Isabel Jalón-Rojas and critically reviewed by all other authors. All authors approved the final version of the paper for submission.

**Competing interests**

The authors declare that they have no conflict of interest.

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

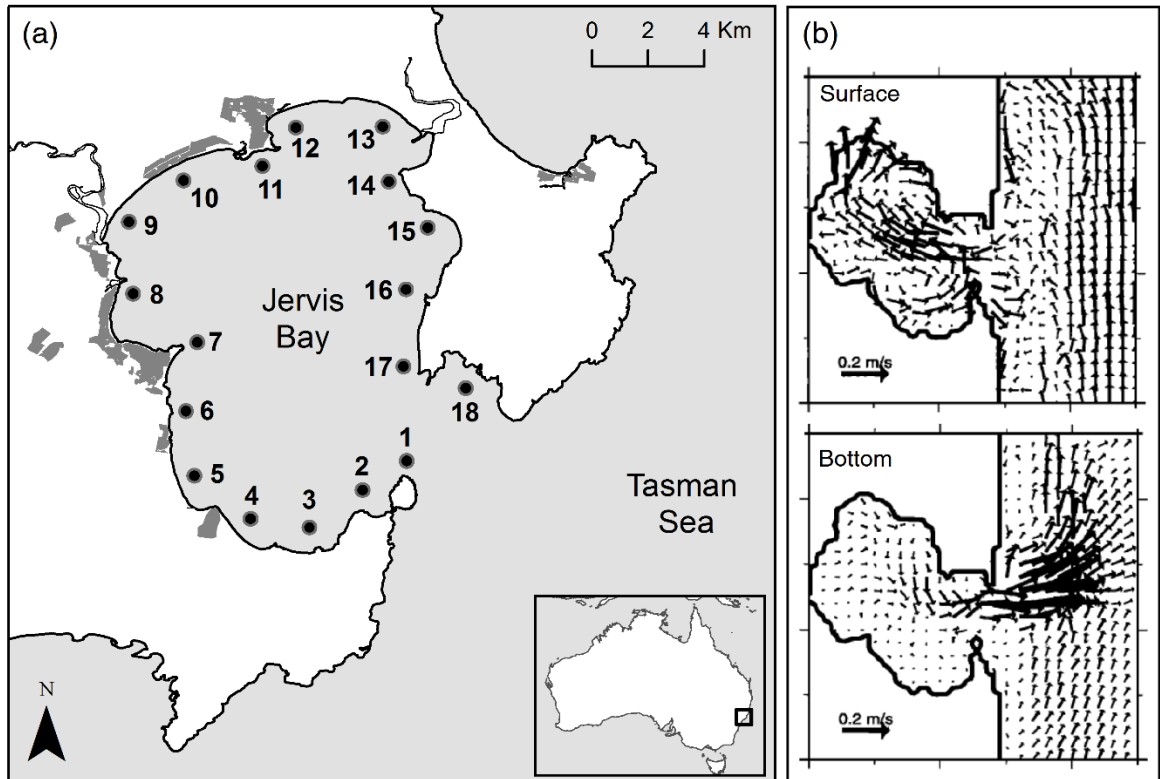

**Figure 1: (a) Map of Jervis Bay (SE Australia). Black dots show the seeding locations and also the centre of destination regions used in the coastal connectivity analysis. Shaded areas represent urban zones. (b) Typical surface and bottom currents of the bay (modified from Wang and Symonds, 1999).**

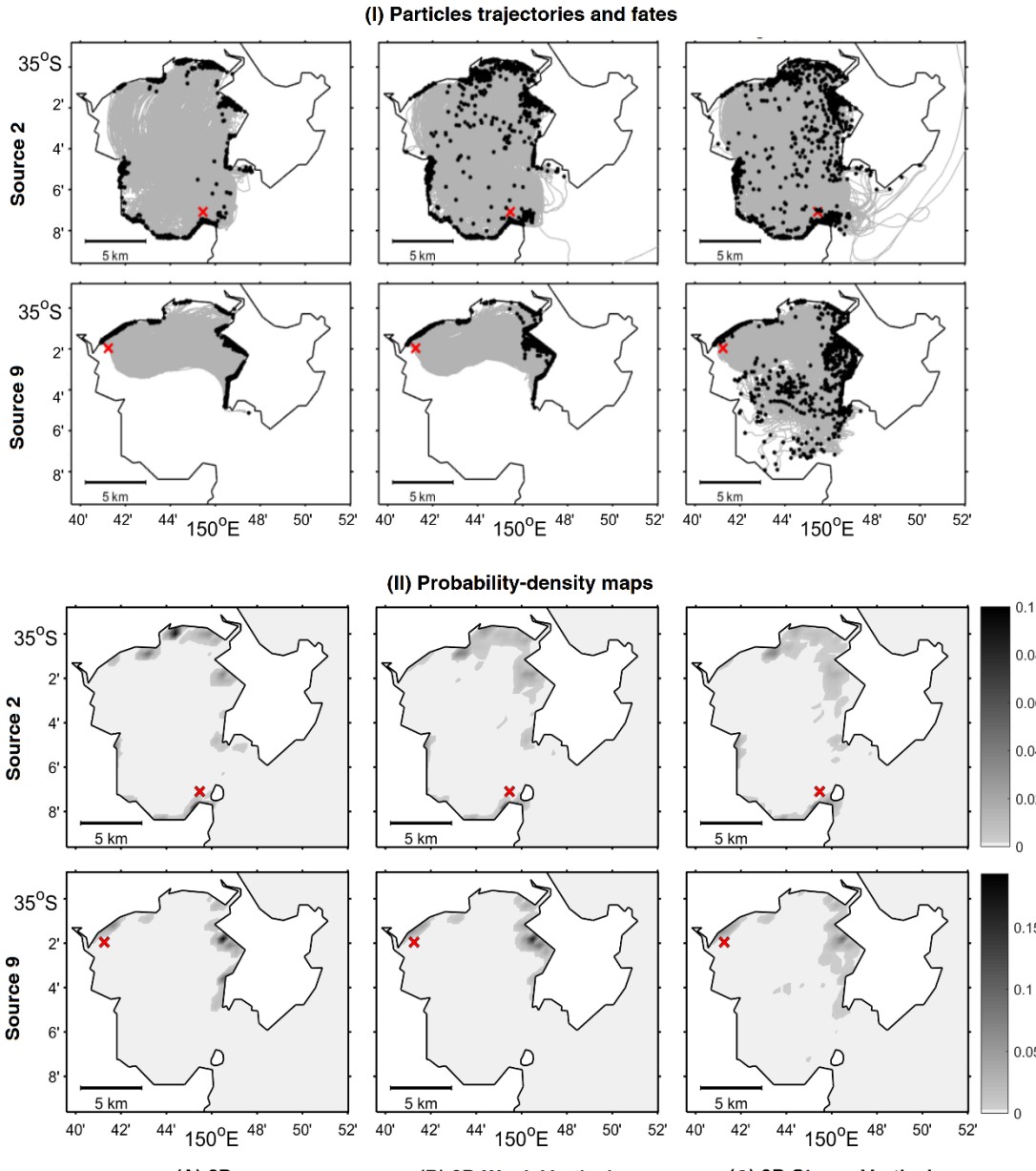

**Figure 2: (I)** Trajectories (grey lines), fate (black dots) and **(II)** probability-density distribution (greyscale bar) of microplastics released at sources 2 and 9 (red crosses) after 5 days of simulation for the three different scenarios: **(A)** 2D approach; **(B)** 3D approach with weak vertical turbulence; **(C)** 3D approach with strong vertical turbulence.

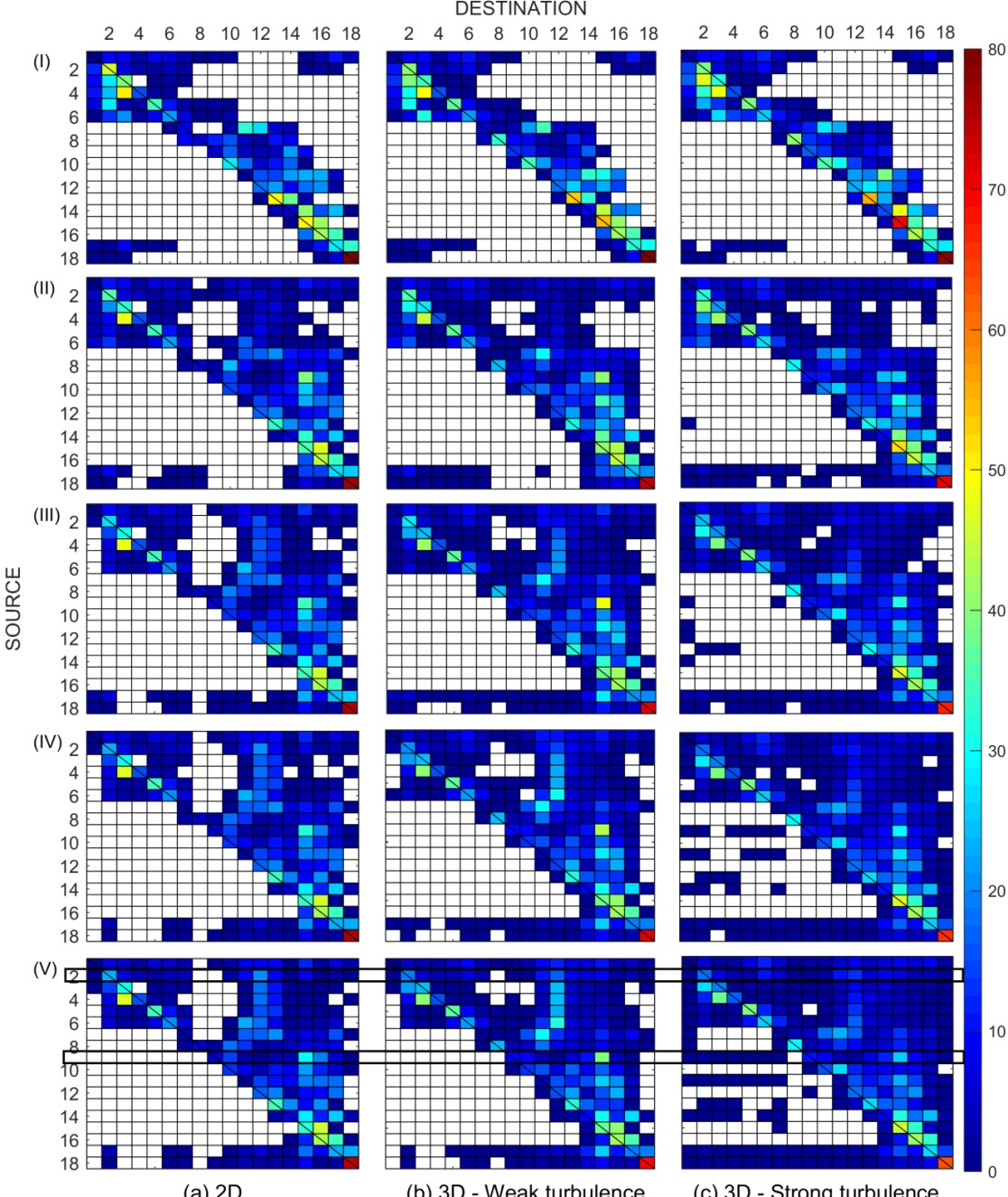

**Figure 3: Coastal connectivity between 18 coastal regions in Jervis Bay (Fig. 1) for simulation times: (I) 1 day; (II) 2 days; (III) 3 days; (IV) 4 days; and (IV) 5 days, and for the three different scenarios: (A) 2D approach; (B) 3D approach with weak vertical turbulence; and (C) 3D approach with strong vertical turbulence. Each matrix shows the percentage of particles released at source j (vertical axis) that travel to destination i (horizontal axis) for a given simulation time. Black-outlined squares in the lower panels (V) highlight the connectivity for the sources and simulation times in Figs. 2.**