# Peer review of "Technical note. On the importance of a three-dimensional approach for modelling the transport of neustic microplastics"

_Ocean Science, 2018_

## Referee Comment (RC1) · Anonymous Referee #1 · 11 Feb 2019

The authors present a comparison of a 2D with 3D microplastic offline transport modelling approach to demonstrate the importance of applying a 3D model. Hydrodynamical forcing from the Princeton Ocean Model (POM) for the period 24 June to 11 July 1998 is used. The paper addresses an actual and interesting issue. The following questions should be clarified:

• What is the spatial and temporal resolution of POM ? What is more or less the size of the first sigma layer for the 2D approach and how does it compare to the mentioned 5m layer thickness (page 2, line 7) ? Assuming the layer is narrow (<5m), does the results differ if you average over the first two or three surface

**C1**

near layers ? How does the 2D layer thickness compare to the average particle depth from the 3D approaches (0.7 and 2.5 m, page 3, line 17/18) ?

- On page 3, line 4 it is written that your aim is not to discuss typical patterns of the microplastic transport and sinking in Jervis Bay. To which extent are your findings transferable to other regions ? Do you think that under certain circumstances a 2D approach could be sufficient ?
- How are the hydrodynamic conditions inside the Bay ? Is it possible to find different periods with different hydrodynamic conditions (stratified, mixed) to generalize more the findings ?
- Do you have informations about the turbulence from POM ? How does it compare to the vertical diffusivity coefficients used for the transport model.
- The density, the size, the shape and the buoyancy of the particles do not go into the study. Can you discuss this point in how far this influences the results ? Microplastic contains a large variety of substances and shapes ?
- Waves are not mentioned. Do you have an idea of its impact and how it compares to the demonstrated differences of a 2D and the 3D approaches ?
- Page 6, line 1: How do you justify your statement that a 3D approach can improve the accuracy ? You see from your study the different outcomes of the different setups, but not how they compare to reality. Particle physical properties (page 5, line35) are not taken into account.

---

## Author Comment (AC1) · 26 Feb 2019

We would like to thank Referee #1 for the interest in our work and the effort spent on reviewing our manuscript. The insightful comments are highly appreciated. In this interactive discussion, we address the general comments; a new version of the manuscript will be submitted with the final revision.

Comment 1. What is the spatial and temporal resolution of POM? What is more or less the size of the first sigma layer for the 2D approach and how does it compare to the mentioned 5m layer thickness (page 2, line 7)? Assuming the layer is narrow(

Figure 1. Depths at sigma layers 5 (A) and 12 (B).

**Comment 2. On page 3, line 4 it is written that your aim is not to discuss typical patterns of the microplastic transport and sinking in Jervis Bay. To which extent are your findings transferable to other regions? Do you think that under certain circumstances a2D approach could be sufficient?**

We agree that we can be more specific. In the revised version, we will precise that our study focuses on coastal shallow waters when relevant. For example, the mentioned statement will be modified as (bold): *"the aim of this work is not to discuss the typical patterns of microplastics transport and sinking in Jervis*"

*Bay; it is a case study to explore the implications of a 2D approach on the simulation accuracy of neusticmicroplastics transport* **in coastal shallow waters**"

We already discussed under which main circumstances our findings would be transferable and a 3D approach would be recommended: stratified systems, high turbulence, under upwelling and downwelling conditions, when simulating non-buoyant particles.

However, we will elongate this discussion and give more details in the revised version:

- We can discuss that our results are transferable to other stratified coastal systems such as estuaries characterized by a density circulation.

- Even if our study focuses on coastal shallow water, surface oceanic water are also characterized by vertical current shear (e.g. wind and wave-driven Ekman flow, density-driven processes, Lund et al., 2015; Lanotte et al., 2016) that could influence the trajectories and final fate of microplastics. Only a 3D approach can consider vertical current shear. A 2D approach could be sufficient when vertical current shear is negligible.

**Comment 3. How are the hydrodynamic conditions inside the Bay? Is it possible to find different periods with different hydrodynamic conditions (stratified, mixed) to generalize more the findings?**

The hydrodynamic conditions inside the bay were described in page 3: "During this simulation period, Jervis Bay was characterized by its typical circulation pattern: clockwise and anticlockwise circulation in the northern and southern regions, respectively. The flow exchange through the entrance was highly stratified, with near-surface inflow on the southern side and deeper outflow on the northern side". In the revised version, we will include a subplot showing the hydrodynamic conditions in surface and bottom waters:

---

## Referee Comment (RC2) · Anonymous Referee #2 · 8 Apr 2019

The manuscript describes the trajectory and fate differences of the neustic microplastic in three scenarios, i.e. 2D only, 3D weak vertical turbulence and 3D strong vertical turbulence. And it tries to indicate the importance of the 3D approach. The manuscript is well organized and clearly analyzed all the numerical experiments. I am providing some comments that are required to be considered by authors. 1. The conclusion of the manuscript is obvious. Because of vertical transport, MPs may be trapped and driven by the horizontal currents at difference depth, for numerical models at difference sigma layer. And due to the difference of the horizontal current field at different sigma layer, the trajectories and fates of MPs in 3 scenarios are different. I noticed the vertical transport of MPs is driven by random walks, vertical current and vertical diffusivity.

The manuscript only evaluated the importance of vertical diffusivity, but what is the contribution of the other two factors? 2. What is the vertical resolution of the hydrodynamic model? How could it be if the vertical resolution changes? 3. page 4 line 22-23: not all the particles stay in the bay in the 3D approach with weak vertical turbulence. 4. page 6 line 1-2: Without validation, there is no stand for the author to conclude a "more-accurate" prediction.

---

## Referee Comment (RC3) · Anonymous Referee #3 · 9 Apr 2019

In this study, the authors propose a comparison of dispersal model outputs for the transport of buoyant marine debris in a semi-enclosed bay (Jervis Bay in Australia) under three frameworks: (i) a two-dimensional depth averaged advection scheme at the surface layer, (ii) a three-dimensional advection scheme with "weak" vertical mixing and (iii) a three-dimensional advection scheme with "strong" vertical mixing. Vertical mixing is formulated using a different vertical diffusivity parameter. Horizontal diffusivity is kept equal between frameworks, so the results are only affected by the inclusion of a third dimension and the effect of random walk vertical mixing. The authors present a series of interpretation of their numerical modelling results discussing distribution, connectivity and transport of Lagrangian particles.

[Figure]

Overall this is a well written manuscript with a clear explanation of numerical modelling methods and the figures are very informative. I would recommend this study for publication in Ocean Science after a few minor revisions. Mainly I would be careful in the conclusion drawn by the authors. Since this is purely a numerical modelling study with no ground-truth data and validation of the model, the authors cannot conclude that a 3D approach is "providing more-accurate predictions". What this study shows is that including a three-dimensional component to a dispersal model alters the connectivity between different marine compartment for marine debris transport in coastal areas.

An interesting result is how the "strong" vertical mixing scenario leaks particles offshore which could explain natural sorting of plastic debris in coastal environment. The authors should further discuss this as well as the relation between vertical mixing and characteristics of marine litter (type, size, buoyancy etc..). I don't think the manuscript as well as the title should focus only on microplastics. Some findings of this study could apply for larger "young" object. Vertical diffusivity of marine debris likely changes with its characteristics, thus the comparison between "weak" and "strong" vertical mixing is an evidence of natural filtering for the transport of marine litter offshore. The authors should emphasize this point. Finally, the formulation and the amplitude of particle beaching is not clearly explained in the manuscript and nor is the influence of vertical advection (W velocity). The authors should provide more details on these aspects.
* * *

---

## Author Comment (AC2) · 9 Apr 2019

We would like to thank Referee #2 for the interest in our work and the effort spent on reviewing our manuscript. In this interactive discussion, we address the general comments; a new version of the manuscript will be submitted with the final revision.

Referee comment: The manuscript describes the trajectory and fate differences of the neustic microplastic in three scenarios, i.e. 2D only, 3D weak vertical turbulence and 3D strong vertical turbulence. And it tries to indicate the importance of the 3D approach. The manuscript is well organized and clearly analyzed all the numerical experiments. I am providing some comments that are required to be considered by

**authors.**

Referee comment 1: The conclusion of the manuscript is obvious. Because of vertical transport, MPs may be trapped and driven by the horizontal currents at difference depth, for numerical models at difference sigma layer. And due to the difference of the horizontal current field at different sigma layer, the trajectories and fates of MPs in 3 scenarios are different. I noticed the vertical transport of MPs is driven by random walks, vertical current and vertical diffusivity. The manuscript only evaluated the importance of vertical diffusivity, but what is the contribution of the other two factors?

Response: We think that our conclusion is not so obvious, because practically all the previous numerical studies ignored the potential impact of vertical transport and used a 2D approach to evaluate the horizontal transport of microplastics. Even if the above statements are known, the scientific community has hypothesized that the vertical transport, induced by turbulence in this case, is not so important as to impact the horizontal trajectories and the fate of microplastics. For that reason, we think it is important to pass this message through this technical note. In addition, we not only show the differences between the 2D and 3D approaches, but we also quantified them. We could give a few more details about this quantification in the conclusions of the final version.

The vertical diffusivity is given through the random walk, but it is true that there is also vertical currents. However, the selected period is characterized by negligible vertical currents, so the differences mainly came from vertical dispersion. We will detail this information in the final version. If required, we could evidence that the advective vertical velocity of this specific study case is not highly impacting the differences between the 2D and the 3D approaches for this case study in supplementary material. However, the impact of vertical velocities for different scenarios such as upwelling and atmospheric cooling will be further analyzed in future works.

Referee comment 2. What is the vertical resolution of the hydrodynamic model? How

could it be if the vertical resolution changes?

Response: As discussed in the response  $n^{\circ}$  1 to referee 1, the model uses a total of 21 sigma layers, so the depth represented by a given sigma layer changes significantly over the space as a function of bottom depth. For example, the 5m surface layer thickness is represented by the layers 1-5 near the mouth while it is represented by layers 1-12 in the inner bay. A decrease of the vertical resolution might decrease the differences between the 2D and the 3D approaches in much as this decrease impacts the accurate representation of currents in this stratified system. As we responded to referee 1, we will describe the vertical resolution and its potential impact in the new version, and discuss the differences with the typical setups for deep ocean models.

Referee comment 3: page 4 line 22-23: not all the particles stay in the bay in the 3D approach with weak vertical turbulence.

Response: We described that all the particles stayed in the bay for the 3D approach with weak vertical turbulence because only one particle (among thousands) left the bay, so the probability of particles going out is negligible for this scenario. We prefer to keep this statement, but we may include some clarification if required.

Referee comment 4. page 6 line 1-2: Without validation, there is no stand for the author to conclude a "more-accurate" prediction.

Response: As discussed in the response  $n^{\circ}$  7 to referee 1, we acknowledge that the lack of observations is a shortcoming of this study. Future work is in progress to apply for funding to conduct field work in Jervis Bay in order to validate the 3D model prediction. This study compares the two approaches by considering the 3D approach "as a reference solution" (page 2, line 12), closer to real conditions, and we evaluated the potential consequences of using a 2D approach, the typical approach used in previous studies. We will modify this statement in the revised version by emphasizing the assumptions of this work.

СЗ

---

## Author Comment (AC3) · 7 May 2019

We would like to thank Referee #3 for the interest in our work and the effort spent on reviewing our manuscript. The insightful comments are highly appreciated. In this interactive discussion, we address the general comments; a new version of the manuscript will be submitted with the final revision. We have detailed the replies in a supplement file.

Please also note the supplement to this comment:
https://www.ocean-sci-discuss.net/os-2018-136/os-2018-136-AC3-supplement.pdf

---

## Author Response (AR1)

We want to thank the three referees for the interest in our work and the effort spent on reviewing our manuscript. The insightful comments are highly appreciated. We have considered all their suggestions and comments, and we have made the modifications/corrections. We have detailed the replies when necessary in the letter. The manuscript with the marked correction is uploaded with this letter.

**REFEREE #1**

**Comment 1. What is the spatial and temporal resolution of POM? What is more or less the size of the first sigma layer for the 2D approach and how does it compare to the mentioned 5m layer thickness (page 2, line 7)? Assuming the layer is narrow(<5m), do the results differ if you average over the first two or three surface near layers? How does the 2D layer thickness compare to the average particle depth from the 3D approaches (0.7 and 2.5 m, page 3, line 17/18)?**

The spatial resolution of POM is 500 m around the bay and the temporal resolution is 12 s. All the model details are described in the mentioned reference Sun et al. (2017), but we have included these details in the revised version (page 3, lines 2-3)[1].

The model uses a total of 21 sigma layers, so the depth represented by a given sigma layer changes significantly over the space as a function of bottom depth. This can be observed in Figure 1 for the fifth and twelfth sigma layers (depths higher than 20 m has not been detailed for a clearer representation inside the bay). We can see that the 5m surface layer thickness is represented by the layers 1-5 near the mouth while it is represented by layers 1-12 in the inner bay. So, we cannot average a given number of layers to represent the 0-5m layer thickness, especially if we also take into account the outer bay. In this paper, we are comparing the 3D approach with the typical 2D approach used in many previous papers that considers particles floating in surface water and only uses surface currents. The only realistic way to represent the surface waters in the 2D approach for a shallow system as Jervis Bay is using the first layer. In the revised version, we have included the number of sigma layer and the 2D layer thickness (0.08 m in the inner bay and 0.3 m at the mouth) (page 3, line 18).

[Figure]

**Figure 1.** Depths at sigma layers 5 (A) and 12 (B).
* * *
[1] Page and line numbers refer to the manuscript with the marked changes.

**Comment 2. On page 3, line 4 it is written that your aim is not to discuss typical patterns of the microplastic transport and sinking in Jervis Bay. To which extent are your findings transferable to other regions? Do you think that under certain circumstances a 2D approach could be sufficient?**

We agree that we can be more specific. In the revised version, we have detailed that our study focuses on coastal shallow waters when relevant (page 2, line 11; page 3, line 9). For example, the mentioned statement will be modified as (bold): "*the aim of this work is not to discuss the typical patterns of microplastics transport and sinking in Jervis Bay; it is a case study to explore the implications of a 2D approach on the simulation accuracy of neustic-microplastics transport **in coastal shallow waters***"

We already discussed under which main circumstances our findings would be transferable and a 3D approach would be recommended: stratified systems, high turbulence, under upwelling and downwelling conditions, when simulating non-buoyant particles.

However, we have elongated this discussion and give more details in the revised version (page 6, lines 1-16):

- Our results can be transferable to other stratified coastal systems such as estuaries characterized by a density circulation.

- Even if our study focuses on coastal shallow water, surface oceanic water are also characterized by vertical current shear (e.g. wind and wave-driven Ekman flow, density-driven processes, Lund et al., 2015; Lanotte et al., 2016) that could influence the trajectories and final fate of microplastics. Only a 3D approach can consider vertical current shear. A 2D approach could be sufficient when vertical current shear is negligible.

**Comment 3. How are the hydrodynamic conditions inside the Bay? Is it possible to find different periods with different hydrodynamic conditions (stratified, mixed) to generalize more the findings?**

The hydrodynamic conditions inside the bay were described in page 3: "*During this simulation period, Jervis Bay was characterized by its typical circulation pattern: clockwise and anticlockwise circulation in the northern and southern regions, respectively. The flow exchange through the entrance was highly stratified, with near-surface inflow on the southern side and deeper outflow on the northern side*". In the revised version, we have included a subplot showing the hydrodynamic conditions in surface and bottom waters:

[Figure]

This is thus the typical circulation of the bay. Conditions can change (coastal trapped waves, upwelling, cooling events) but all these processes are baroclinic (e.g. Wang and Symonds, 1999; Sun et al., 2017; Liao and Wang, 2018), so the 2D approach is not suitable in coastal systems such as Jervis Bay. We have included this information in the reviewed version (page 6, lines 1-3). Our conclusions are transferable as articulated in the previous question.

**Comment 4. Do you have information about the turbulence from POM? How does it compare to the vertical diffusivity coefficients used for the transport model?**

The model uses the turbulence closure scheme described by Mellor and Yamada (1982) for vertical mixing coefficients, which is a time variable. The transport model uses the typical constant diffusivity coefficients typically uses in the literature because our objective is not evaluating the real conditions of Jervis Bay but the potential range of conditions that can occur in these environments (page 3, lines 18-20).

**Comment 5. The density, the size, the shape and the buoyancy of the particles do not go into the study. Can you discuss this point in how far this influences the results? Microplastics contains a large variety of substances and shapes?**

The objective of this technical note is to compare the 2D and 3D approaches just for low-dense positive-buoyant neustic microplastics. The motivation is that previous works only modelled the transport of this type of microplastics using a 2D approach. As discussed in the manuscript, our results suggest that "*the vertical movement of particles induced by other physical processes, such as **particle sinking (in the case of non-buoyant particles)**, upwelling and downwelling, could also affect the horizontal transport of microplastics, even in a higher degree, and a 3D approach could be mandatory*" (page 6, lines 8-11). So we already mentioned that we expect that buoyancy has even a higher impact on microplastics trajectories, but we cannot give more details at this point.

However, we also pointed that "*Further progress on microplastics modelling requires thus the development of three-dimensional models that consider the particle sinking, which in turn depends on particle physical properties (density, size, shape, Chubarenko et al., 2016)*" (page 6, lines 13-15). And this is effectively what we have done. Based on the conclusions of this technical note, we have developed a 3D model that considers the influence of these three physical properties, but also of biofilm properties and other physical processes such as washing off from the beach. The model description and the discussion of the relative impact of each property/process are the objectives of our next paper that has been recently published (*Jalon-Rojas et al. A 3D numerical model to Track Marine Plastic Debris (TrackMPD): Sensitivity of microplastics trajectories and fates to particle dynamical properties and physical processes, Marine Pollution Bulletin*). We have included this reference in the technical note (page 6, lines 15-16).

**Comment 6. Waves are not mentioned. Do you have an idea of its impact and how it compares to the demonstrated differences of a 2D and the 3D approaches?**

The impact of waves on microplastics transport is a different subject of study (which we intend to conduct in near future). However, when we discuss the transferability of our results, we have also mentioned that waves enhance vertical mixing (e.g. Deepwell, and Stastna, 2016) and may also impact the vertical displacement of particles near the surface (page 6, line 9).

**Comment 7. Page 6, line 1: How do you justify your statement that a 3D approach can improve the accuracy?  You see from your study the different outcomes of the different setups, but not how they compare to reality. Particle physical properties (page 5, line 35) are not taken into account.**

All the reviewers made the same comment and we agree with them. We acknowledge that the lack of observations is a shortcoming of this study. Future work is in progress to apply for funding to conduct field work in Jervis Bay in order to validate the 3D model prediction. This study compares the two approaches by considering the 3D approach "as a reference solution" (page 2, line 13), closer to real conditions, and we evaluated the potential consequences of using a 2D approach, the typical approach used in previous studies. We have modified this sentence as suggested by other reviewer: "providing more accurate predictions" was modified by "which impact the predictions" (pag 6, line 18).

As discussed in Comment 5, we expected that the 3d approach will be even more important for negative-buoyant particles and this result has motivated a new study has been recently published.

**REFEREE #2**

**Comment 1.  The conclusion of the manuscript is obvious.  Because of vertical transport, MPs may be trapped and driven by the horizontal currents at difference depth, for numerical models at difference sigma layer.  And due to the difference of the horizontal current field at different sigma layer, the trajectories and fates of MPs in 3 scenarios are different.  I noticed the vertical transport of MPs is driven by random walks, vertical current and vertical diffusivity.  The manuscript only evaluated the importance of vertical diffusivity, but what is the contribution of the other two factors?**

We think that our conclusion is not so obvious, because practically all the previous numerical studies (see references in the manuscript, page 2, lines 1-2) ignored the potential impact of vertical transport and used a 2D approach to evaluate the horizontal transport of microplastics.  Even if the above statements

are known, the scientific community has hypothesized that the vertical transport, induced by turbulence in this case, is not so important as to impact the horizontal trajectories and the fate of microplastics. For that reason, we think it is important to pass this message through this technical note. In addition, we not only show the differences between the 2D and 3D approaches, but we also quantified them for an specific case study.

The vertical diffusivity is given through the random walk, but it is true that there are also vertical currents. However, the selected period is characterized by negligible vertical currents (3-4 order of magnitude lower to horizontal currents), so the differences mainly came from vertical dispersion. We have detailed this information in the final version (page 3, lines 6-7). In addition, we have proved that the differences between the 2D and 3D scenarios are mainly due to vertical dispersion by comparing the probability density map of two scenarios: (a) 3D approach with low turbulent conditions and vertical currents; (b) 3D with low turbulent conditions and no vertical currents (see Figure below). Results show that the accumulation patterns of particles are practically identical for both scenarios, so vertical velocities have a low impact and the difference between the 2D and 3D approaches discussed in this paper, are mainly related to vertical dispersion.

[Figure]

This figure has been included in Supplementary Material 2.

**Comment 2. What is the vertical resolution of the hydrodynamic model? How could it be if the vertical resolution changes?**

As discussed in the response n∘1 to referee 1, the model uses a total of 21 sigma layers, so the depth represented by a given sigma layer changes significantly over the space as a function of bottom depth. For example, the 5m surface layer thickness is represented by the layers 1-5 near the mouth while it is represented by layers 1-12 in the inner bay. In the revised version, we have included the vertical resolution of the first layer in the inner bay and near to the mouth (page 3, line 18)

A decrease of the vertical resolution might decrease the differences between the 2D and the 3D approaches in much as this decrease impacts the accurate representation of currents at the different vertical layers in the system. We have included a sentence to highlight the importance of vertical resolution (page 5, lines 33-34).

**Comment 3. page 4 line 22-23: not all the particles stay in the bay in the 3Dapproach with weak vertical turbulence.**

We described that all the particles stayed in the bay for the 3D approach with weak vertical turbulence because only one particle (among thousands) left the bay, so the probability of particles going out is negligible for this scenario. We prefer to keep this statement, but we may include some clarification if required.

**Referee comment 4. page 6 line 1-2: Without validation, there is no stand for the author to conclude a "more-accurate" prediction.**

See the response n∘7 to referee 1.

**REFEREE #3**

**Comment 1. Mainly I would be careful in the conclusion drawn by the authors. Since this is purely a numerical modelling study with no ground-truth data and validation of the model, the authors cannot conclude that a 3D approach is "providing more-accurate predictions". What this study shows is that including a three-dimensional component to a dispersal model alters the connectivity between different marine compartment for marine debris transport in coastal areas.**

See the response n∘7 to referee 1; the conclusion has been modified as suggested by the reviewer (page 6, line 18).

**Comment 2. An interesting result is how the "strong" vertical mixing scenario leaks particles off-shore which could explain natural sorting of plastic debris in coastal environment. The authors should further discuss this as well as the relation between vertical mixing and characteristics of marine litter (type, size, buoyancy etc..).**

We already discussed the role of vertical mixing on the "scape" of particles from the bay (page 4, lines 25-28). As discussed at the response n∘5 to referee 1, the objective of this paper is to discuss the relevance of a 2D approach for floating low-dense particles, and the impact of the physical properties of microplastics have been discussed in depth in our next paper (reference included in the revised version: page 6, lines 15-16).

**Comment 3. I don't think the manuscript as well as the title should focus only on microplastics. Some findings of this study could apply for larger "young" object.**

We prefer to focus the paper on microplastics since the initial motivation is that most of the studies on microplastic modelling consider a 2D approach. However, we have included that the finding can also be applied for other floating objects (page 6, lines 5-6).

**Comment 4. Vertical diffusivity of marine debris likely changes with its characteristics, thus the comparison between "weak" and "strong" vertical mixing is an evidence of natural filtering for the transport of marine litter offshore. The authors should emphasize this point.**

This paper focuses on the modelling of one kind of particles: low-dense floating microplastics. We have briefly discussed the implications of these results for another kind of particles, but this is the object of our following paper (Jalon-Rojas et al. 2019, Marine Pollution Bulletin).

**Comment 5. Finally, the formulation and the amplitude of particle beaching is not clearly explained in the manuscript and nor is the influence of vertical advection (W velocity). The authors should provide more details on these aspects.**

We have detailed the beaching in the revised version (page 2, lines 24-25). Regarding vertical advection, see the response n◦1 to referee 2.

[revised manuscript text omitted]